# Neural signatures of perceptual inference

William Sedley[1,2]*, Phillip E Gander[2], Sukhbinder Kumar[1,2,3],
Christopher K Kovach[2], Hiroyuki Oya[2], Hiroto Kawasaki[2], Matthew A Howard III[2],
Timothy D Griffiths[1,2,3]

[1]Institute of Neuroscience, Newcastle University, Newcastle upon Tyne, United
Kingdom; [2]Human Brain Research Laboratory, University of Iowa, Iowa, United
States; [3]Wellcome Trust Centre for Neuroimaging, University College London,
London, United Kingdom

**Abstract** Generative models, such as *predictive coding*, posit that perception results from a combination of sensory input and prior *prediction*, each weighted by its *precision* (inverse variance), with incongruence between these termed *prediction error* (deviation from prediction) or *surprise* (negative log probability of the sensory input). However, direct evidence for such a system, and the physiological basis of its computations, is lacking. Using an auditory stimulus whose pitch value changed according to specific rules, we controlled and separated the three key computational variables underlying perception, and discovered, using direct recordings from human auditory cortex, that *surprise* due to prediction violations is encoded by local field potential oscillations in the gamma band (>30 Hz), changes to *predictions* in the beta band (12-30 Hz), and that the *precision* of predictions appears to quantitatively relate to alpha band oscillations (8-12 Hz). These results confirm oscillatory codes for critical aspects of generative models of perception.

## Introduction

It has long been apparent that brain responses do not simply represent input from sensory organs, but that they are modulated by context and expectation, giving rise to phenomena such as priming, mismatch negativity and repetition suppression. These can be explained if perceptual systems are based on internal generative models of the environment which are constantly updated based on experience. Predictive coding (*Rao and Ballard, 1999*) is a popular account of perception, in which internal representations generate *predictions* about upcoming sensory input, characterised by their mean and *precision* (inverse variance) (*Friston, 2005*; *Friston and Kiebel, 2009*). Sensory information is processed hierarchically, with backward connections conveying predictions, and forward connections conveying violations of these predictions, namely *prediction errors*. Qualitatively, prediction errors are the mismatch between the prediction and incoming sensory information, but the term is often used without a quantitative definition. One quantitative formulation of prediction error is *surprise* (*Friston and Kiebel, 2009*), which is the negative log probability of a sensory event, given the prior prediction. This definition takes into account the *precision* of predictions, such that the same prediction violation causes greater surprise where predictions are more precise. Prediction errors act to produce changes in predictions, thereby updating and refining internal models of the environment, and reducing subsequent prediction errors. These variables are illustrated in *Figure 1*. There is substantial overlap between predictive coding and other accounts of perception based on internal generative models (*Friston, 2008*). The crucial common feature of any generative model of perception is the brain's use of hidden states to predict observed sensory inputs, thus the methods and findings of this study are applicable to all generative perceptual models.

The functional unit of neocortex, the *canonical microcircuit* (*Haeusler and Maass, 2007*), has recently been interpreted in light of predictive coding models (*Bastos et al., 2012*), revealing

*For correspondence: willsedley@ gmail.com

Competing interests: The authors declare that no competing interests exist.

**eLife digest** Our perception of the world is not only based on input from our senses. Instead, what we perceive is also heavily altered by the context of what is being sensed and our expectations about it. Some researchers have suggested that perception results from combining information from our senses and our predictions. This school of thought, referred to as "predictive coding", essentially proposed that the brain stores a model of the world and weighs it up against information from our senses in order to determine what we perceive.

Nevertheless, direct evidence for the brain working in this way was still missing. While neuroscientists had seen the brain respond when there was a mismatch between an expectation and incoming sensory information, no one has observed the predictions themselves within the brain.

Sedley et al. now provide such direct evidence for predictions about upcoming sensory information, by directly recording the electrical activity in the brains of human volunteers who were undergoing surgery for epilepsy. The experiment made use of a new method in which the volunteers listened to a sequence of sounds that was semi-predictable. That is to say that, at first, the volunteers heard a selection of similarly pitched sounds. After random intervals, the average pitch of these sounds changed and they became more or less variable for a while before randomly changing again. This approach meant that the volunteers had to continually update their predictions throughout the experiment

In keeping with previous studies, the unexpected sounds, which caused a mismatch between the sensory information and the brain's prediction, were linked to high-frequency brainwaves. However, Sedley et al. discovered that updating the predictions themselves was linked to middle-frequency brainwaves; this confirms what the predictive coding model had suggested. Finally, this approach also unexpectedly revealed that how confident the volunteer was about the prediction was linked to low-frequency brainwaves.

In the future, this new method will provide an easy way of directly studying elements of perception in humans and, since the experiments do not require complex learning, in other animals too.

appropriate neuronal properties and internal/external connectivity to carry out the necessary neuronal computations. It is thus hypothesised that superficial cell populations calculate prediction errors, manifest as gamma-band oscillations (>30 Hz), and pass these to higher brain areas, while deep cell populations encode predictions, which manifest as beta band oscillations (12–30 Hz) and pass these to lower brain areas (*Bastos et al., 2012*). The layer-specific separation of higher and lower frequency oscillations (*Spaak et al., 2012*), and the forward/backward asymmetry of high/low frequency oscillations (*Buschman and Miller, 2007*; *Fontolan et al., 2014*; *van Kerkoerle et al., 2014*, *Bastos et al., 2015*), are supported by direct evidence. A number of studies have found oscillatory gamma magnitude to correlate with the unexpectedness of incongruence of stimuli (*Arnal et al., 2011*; *Brodski et al., 2015*; *Todorovic et al., 2011*), but it remains unclear exactly what computational variable they represent. While there is a strong case that beta oscillations are involved in top-down neural communication, evidence specifically linking beta oscillations to predictions is presently limited and indirect (*Arnal and Giraud, 2012*), but includes observations that there is interdependence of gamma and subsequent beta activity in both in vivo (*Haenschel et al., 2000*) and in silico (*Kopell et al., 2011*) studies and that omissions of expected stimuli induce a beta rebound response (*Fujioka et al., 2009*). An oscillatory correlate of precision, to our knowledge, has not been proposed, though precision might affect the magnitude of gamma responses to prediction violations (*Feldman and Friston, 2010*). While an oscillatory correlate is possible, a case has been made that neuromodulatory connections alone, for instance from the basal forebrain cholinergic system, may be sufficient to dynamically mediate precision in sensory hierarchies (*Feldman and Friston, 2010*; *Kanai et al., 2015*). Direct evidence for correlates of processes inherent in perceptual inference requires being able to quantitatively manipulate predictions during an experiment, which has not so far been achieved.

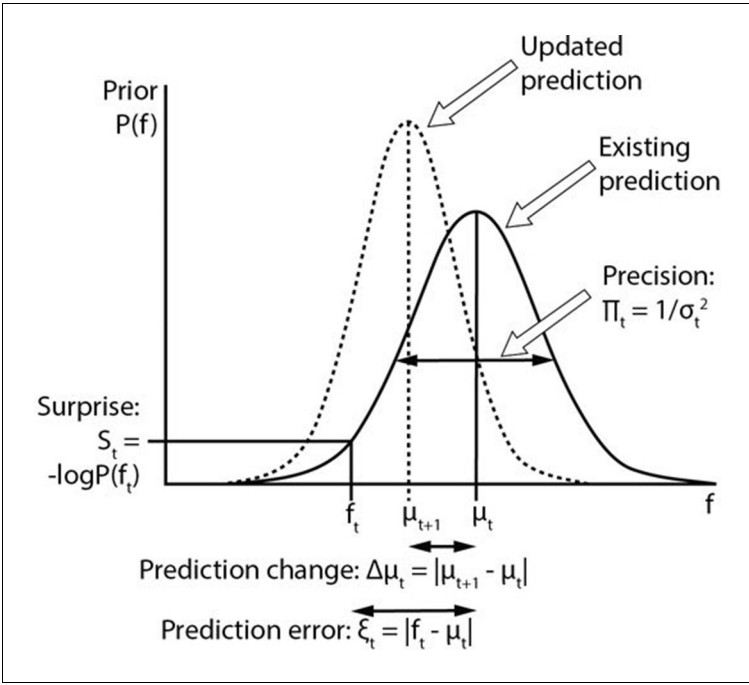

**Figure 1.** Computational variables involved in perceptual inference. The graph displays a schematic probability distribution (solid curve) representing the prior prediction about the fundamental frequency (f) of an upcoming auditory stimulus ($f_t$), where t simply refers to the number or position of the stimulus within a sequence. This prediction is characterised by its mean ($\mu_t$) and precision ($\Pi_t$), which is the inverse of its variance ($\sigma^2$). The incongruence between the actual $f_t$ and the prediction can be expressed either as a (non-precision-weighted) prediction error ($\xi_t$), that is, the absolute difference from the prediction mean, or as surprise ($S_t$), that is, the negative log probability of the actual $f_t$ value according to the prediction distribution. As a result of a mismatch with bottom up sensory information, the prediction changes (dashed line). The change to the prediction ($\Delta\mu_t$) is calculated simply as the absolute difference between the old ($\mu_t$) and new ($\mu_{t+1}$) prediction means. Note that the curves on the graph display changing predictions on account of a stimulus (i.e. Bayesian belief updating) as opposed to the more commonly encountered graph in this field of research where the curves indicate the prior prediction, the sensory information and the posterior inference about the individual stimulus (i.e. Bayesian inference).

In the present study, we sought to dissociate and expose the neural signatures of four key variables in predictive coding and other generative accounts of perception, namely *surprise, prediction error, prediction change* and *prediction precision*. Here, *prediction error* refers to absolute deviation of a sensory event from the mean of the prior prediction (which does not take into account the precision of the prediction). We hypothesised that surprise (over and above prediction error) would correlate with gamma oscillations, and prediction change with beta oscillations. The possibility of an oscillatory code for precision was also explored.

## Results and discussion

Direct cortical recordings were made from the auditory cortices of three awake humans undergoing invasive monitoring for epilepsy localization, while they listened to a pitch stimulus with a fundamental frequency (usually referred to as 'f0'; hereafter just 'f' for clarity) that varied according to simple rules (*Figure 2*). Local field potential (LFP) data were decomposed using Morlet wavelets, separated into evoked and induced components, and regressed against the four perceptual inference variables of interest which were calculated by Bayes-optimal inversion of the sequence of f values assuming full knowledge of the rules by which they were generated (*Figure 2—figure supplements 1* and *2*).

In keeping with prior hypotheses, both *surprise* and *prediction error* (the latter not taking into account the precision of predictions) were associated with significant gamma band responses in the

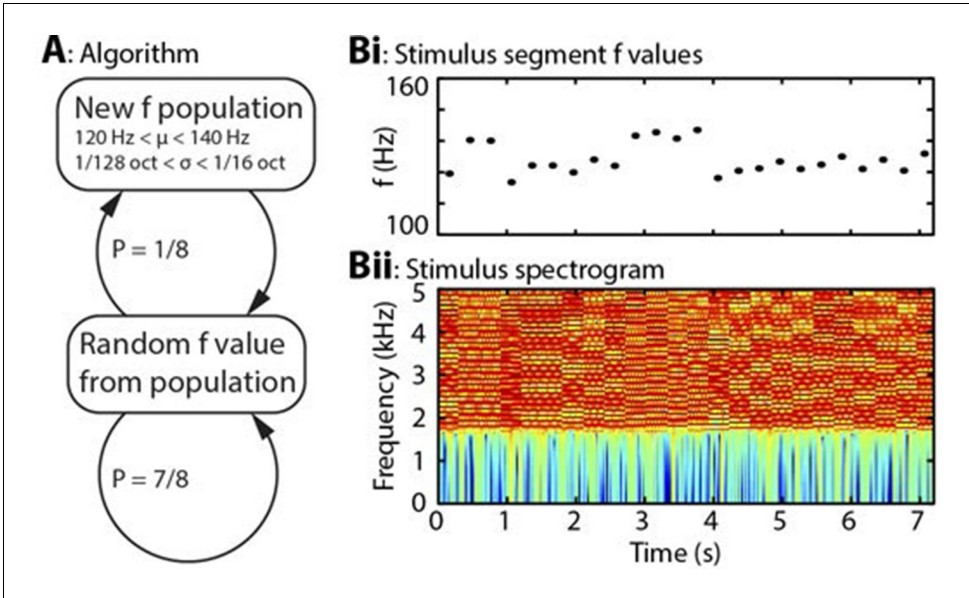

**Figure 2.** Algorithm and example stimulus. (**A**) The stimulus is composed of a series of concatenated segments, differing only in fundamental frequency (f). At any time, a given f population is in effect, characterised by its mean (μ) and standard deviation (σ). For each successive segment, there is a 7/8 chance that that segment's f value will be randomly drawn from the present population, and a 1/8 chance that the present population will be replaced, with new μ and σ values drawn from uniform distributions. (**B**) Example section of stimulus. (**Bi**) Dots indicate the f values of individual stimulus segments, of 300 ms duration each. Four population changes are apparent. (**Bii**) Spectrogram of the corresponding stimulus, up to 5 kHz, on a colour scale of -60 to 0 dB relative to the maximum power value. The stimulus power spectrum does not change between segments, and the only difference is the spacing of the harmonics.

The following figure supplements are available for figure 2:

**Figure supplement 1.** Generative model and inversion scheme.

**Figure supplement 2.** Example Bayes-optimal prior predictions generated by model inversion.

**Figure supplement 3.** Regressor correlations.

LFP. We first established which of these variables explained the LFP data better. *Figure 3* shows the strong correlation between these variables (A), the explanatory power of each with respect to the LFP data (B), and the unique explanatory power of each after partialling out the other variable (C). The extremely strong correlation between *surprise* and *prediction error* (r = 0.92 over 8000 samples) necessitated this partial analysis (C) in order to examine the independent contribution of each variable to the observed LFP data. Both variables correlated positively with gamma magnitude, but *surprise* showed a stronger correlation in all three subjects. In the partial analysis (C), residual *surprise* (after partialling out *prediction error*) correlated positively with gamma magnitude, whereas residual *prediction error* (after partialling out *surprise*) showed only a weak negative correlation in two subjects, and no correlation in one subject. At group level, these correlations were significantly different to each other at p<0.01 corrected, thus we concluded that *surprise* is the better correlate of gamma magnitude, and used this measure for further analysis.

*Figure 4* shows, at group level, the spectrotemporal pattern of induced and evoked oscillations uniquely attributable to each of the three perceptual variables of interest: surprise (S), change in prediction mean (Δμ), precision of predictions (Π), as well as the change in f value from one stimulus to the next (Δf). The latter measure was not a perceptual variable of interest but was included for comparative purposes as it approximately represents the 'pitch onset response' which is a robust and familiar response in auditory neurophysiology (*Griffiths et al., 2010*). Data significant at p<0.05

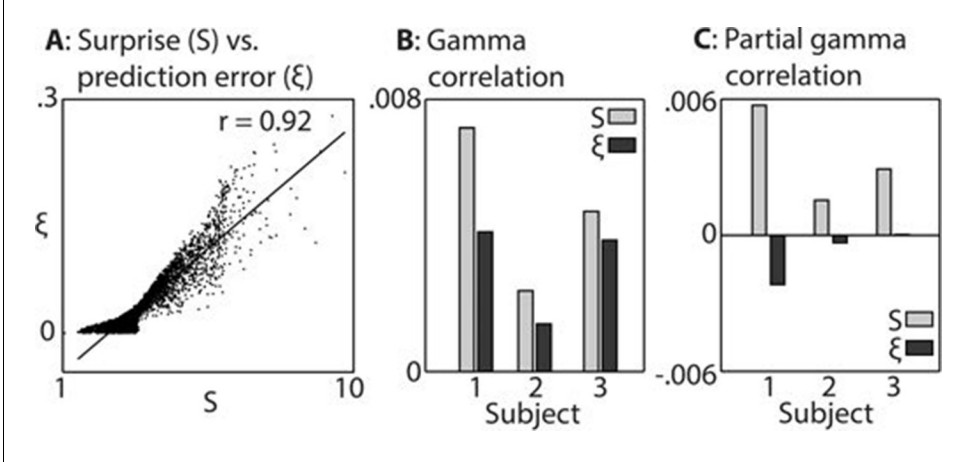

**Figure 3.** Comparison between surprise and prediction error. (**A**) Correlation between surprise (S) and non-precision-weighted prediction error ($\xi$), with each dot indicating an individual stimulus segment and the line indicating a linear regression fit. (**B/C**) Mean Pearson product moment correlation coefficients (r) between $S_t$ or $\xi_t$, and gamma oscillation magnitude (30–100 Hz) in the 90–500 ms period following the onset of stimulus segment t. Regression coefficients were calculated for each time-frequency point, after partialling out the influences of current and preceding/subsequent values of all other regressors, and then averaged across time and frequency; these processing steps diminished the absolute size of the correlation values. In C, the influence of S on $\xi$, and $\xi$ on S, was also partialled out, thus exposing the unique contribution of each variable to explaining the observed neural response. Partial S showed a higher mean correlation, across subjects, with gamma magnitude than partial $\xi$ (p<0.01).

corrected (based on a non-parametric permutation approach [*Maris and Oostenveld, 2007*]) are shown in the left column of each group, and all data in the right column. Individual subject data are shown in *Figure 4—figure supplement 1*. As these perceptual variables were highly correlated (*Figure 2—figure supplement 3*), instantaneously and over time, regression for the main analysis was based on the residuals after partialling out these correlated influences, such that only the unique explanatory contribution of each variable, with respect to the LFP data, was measured. While the correlation values observed were small in absolute terms (Pearson's r < 0.1), reassuringly these r values were of the same scale of magnitude as those for change in frequency ($\Delta f$), which represents a robust auditory response. Furthermore, the LFP data variance explained by the entire model (*Figure 4—figure supplement 1*) was around 1%. In accordance with our hypotheses, *surprise* (S) correlated positively, across subjects, with gamma oscillations, beginning at around 100 ms from segment onset, and this was significant by 200 ms. Also in accordance with our hypothesis, changes to *predictions* ($\Delta\mu$) correlated positively with beta oscillations coinciding with the onset of the subsequent stimulus segment (about 100 ms after), which again was significant. Prediction *precision* ($\Pi$) correlated positively with delta-alpha (2–12 Hz) frequency oscillations (for the whole 0–300 ms period from segment onset), although this was only significant in the alpha frequency range, and fell slightly below significance in the delta-theta range. Given the strong negative correlation between $\Pi$ and the preceding values of S and $\Delta f$, it seemed likely that the low-frequency correlates of these were being mutually attenuated by the partialisation process. For this reason, and to search for correlates of the commonalities between key variables, we repeated the analyses with only the contemporaneous value of $\Delta f$ being partialled out (*Figure 4—figure supplement 3*). This analysis found highly significant correlates of *precision* ($\Pi$) in the full delta-alpha range, spanning the previous, current and subsequent segments, but we cannot attribute the delta-theta component to $\Pi$ with absolute confidence. To respect the exploratory nature of our search for oscillatory correlates of $\Pi$, a further variation of the analysis (*Figure 4—figure supplement 4*) omitted $\Pi$ altogether (including the partialisation of other variables with respect to it). Results were quantitatively stronger (due to reduced partialisation), but qualitatively similar except that S contained a strong negative delta-alpha band correlation coincident with the subsequent stimulus segment. As S for one segment is negatively correlated with $\Pi$ of the subsequent segment, it is thus not presently clear how much of

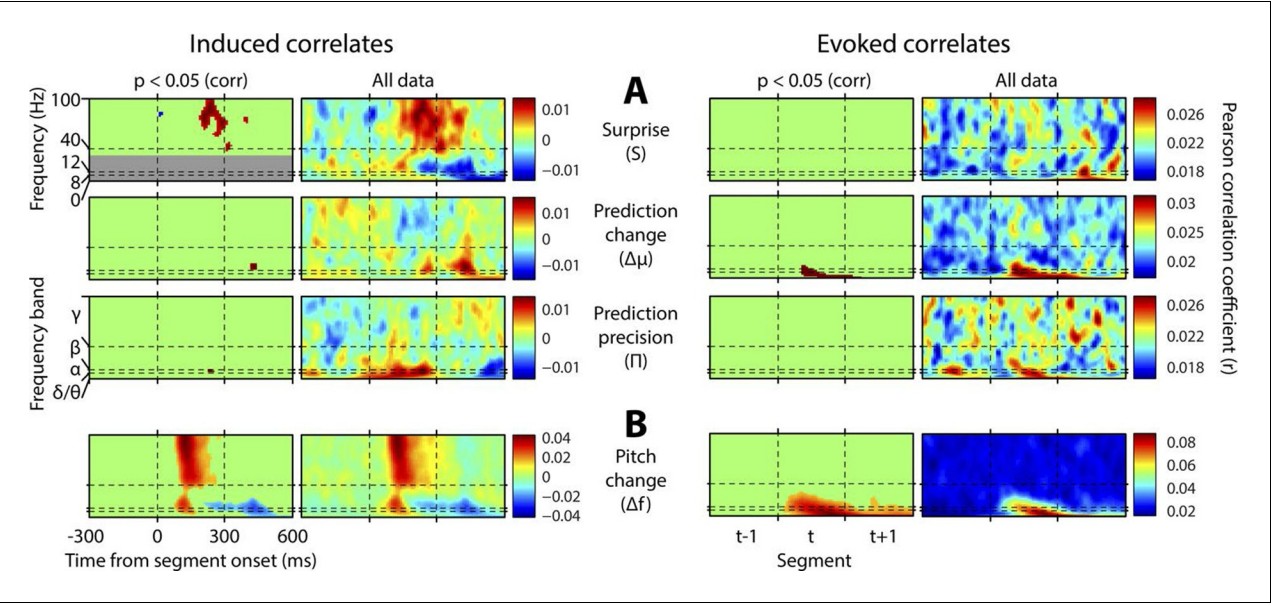

**Figure 4.** Spectrotemporal profiles associated with key perceptual inference variables. Each plot illustrates the mean Pearson product moment correlation coefficient (r), across stimulus-responsive electrodes and across subjects, between induced oscillatory amplitude, at each time-frequency point, and the regressor of interest, that is, a time-frequency 'image' of the oscillatory correlates of a particular perceptual variable. Time is represented on horizontal axes, and frequency on vertical. Dashed lines indicate the division between the previous (t-1), current (t) and subsequent (t+1) stimulus segment (vertical lines), and between frequency bands (horizontal lines). Each row of plots represents one regressor. The left-hand group contains induced correlates, and the right-hand group evoked, with the left-hand column in each group showing data points significant at p<0.05 corrected. Regressors are partialised with respect to each other, such that only the unique contribution of each to explaining the overall oscillatory data is displayed. (A) Spectrotemporal correlates of the three fundamental variables for perceptual inference. Note that the grey area in the upper left plot reflects the spectrotemporal region of interest (ROI) analysis used for correlates of surprise (i.e. >30 Hz). Outside of the ROI analysis, no significant correlates were observed below 30 Hz. (B) The variable 'pitch change' indicates the overall response to a changing stimulus and is included to illustrate the magnitude and time-frequency distribution of a typical and robust auditory response. δ = delta (0–4 Hz), θ = theta (4–8 Hz), α = alpha (8–12 Hz), β = beta (12–30 Hz), γ = gamma (30–100 Hz).

The following figure supplements are available for figure 4:

**Figure supplement 1.** Individual subject induced correlates.

**Figure supplement 2.** Total variance explained by the model.

**Figure supplement 3.** Results without mutual partialisation.

**Figure supplement 4.** Results with prediction precision omitted.

**Figure supplement 5.** Electrode positions.

**Figure supplement 6.** Distribution of correlations across electrodes.

this low-frequency correlation is with S (negatively), and how much with Π (positively). However, given its sole significant association in the main analysis (*Figure 4*) is with Π, and that a low-frequency correlate of S is not expected based on prior literature, we favour the interpretation that low-frequency oscillations are a correlate of the precision of prior predictions. Evoked results are shown in the right hand sections of *Figure 4* plus its figure supplements 3 and 4. Unlike the induced results, there was no qualitative distinction between the timing or frequency profiles of the different variables (this is particularly evident in *Figure 4—figure supplement 4*). The only significant evoked responses in the main analysis (*Figure 4*) were to Δf and Δμ, while S only showed significant evoked correlates when Π was omitted from the analysis (*Figure 4—figure supplement 4*).

In light of the above results, this study provides the first direct demonstration that beta oscillations are involved in updating the content of sensory predictions. We have also found that the precision of predictions is correlated to the magnitude of alpha oscillations, and possibly delta and/or theta also, thereby raising the possibility of an oscillatory mechanism for the control of precision. Interestingly, this correlate of precision was not time-locked to the stimuli (evoked), despite delta-theta oscillations showing strong phase entrainment by stimuli when predicting stimulus timing (*Arnal and Giraud, 2012*), and the period of the stimulus segments (300 ms) falling within the delta-theta range. Possibilities include that there is an evoked correlate closely shared by both surprise and precision which the present methods are unable to disambiguate, or that this time-locking of low-frequency oscillations is entrained preferentially by low-level stimulus features (such as changes in stimulus power spectrum) as opposed to the higher level feature of temporal pitch employed in the present study. Existing generative accounts of perception have not proposed a specific oscillatory correlate for the precision of predictions in predicting *what* a sensory stimulus will be. However, the present findings are not without precedent, as theta (4–8 Hz) and alpha (8–12 Hz) oscillations are implicated in mechanisms to predict *when* a stimulus will occur, with theta phase aligning to the expected stimulus onset (*Arnal and Giraud, 2012*), and alpha magnitude has been found to correlate with the probability of a stimulus change occurring (*Bauer et al., 2014*). Both theta (*Canolty et al., 2006*) and alpha (*Jensen and Mazaheri, 2010*) oscillations modulate higher frequency oscillations, through phase-amplitude coupling, and thereby segregating sensory responses into specific temporal windows. There is also antagonism between theta/alpha and beta/gamma oscillation magnitudes (*Spaak et al., 2012*) which, in the context of a theta/alpha code for precision, might indicate that over coarse time scales neuronal populations alternate between states of precise predictions (with theta/alpha predominating) and states of prediction violation (with beta/gamma predominating). However, the present results would suggest more than simple reciprocal antagonism and reveal the specific computational role of each oscillation type. While an oscillatory code for precision could have far-reaching implications, we must respect the fact that this is a novel finding and thus requires corroboration from additional studies with alternative methodology.

Perhaps most strikingly, we have shown that the key variables theoretically necessary for sensory inference have distinct oscillatory profiles, with little to no overlap between these, which show remarkable consistency across subjects. Furthermore, each oscillatory frequency band correlates with a distinct computational variable. Thus, the present findings may be able to retrospectively aid in the interpretation of a large number of studies examining induced oscillations. As generative accounts of perception are generic across stimulus dimensions and sensory modalities, and perhaps even all of brain function (for instance, if action is understood as a method of resolving prediction errors [*Friston et al., 2006*]), the applicability of the results may be very broad indeed. The present paradigm is instantly portable to any sensory modality and, given the absence of any training or task requirement, to any species.

## Materials and methods

### Stimulus algorithm

The basis of the experiment was an algorithm (*Figure 2A*) in which stimulus segments varied across only one perceptual dimension, and values were drawn randomly from *populations*, that is, Gaussian distributions, each characterised by its mean ($\mu$) and standard deviation ($\sigma$). These populations constituted hidden states that were not directly observable, but whose parameters (i.e. $\mu$ and $\sigma$) could be inferred. The populations were randomly changed according to simple rules, such that subjects could be expected to unconsciously learn these rules in order to minimise uncertainty about upcoming stimuli. The rules were that for each stimulus segment, there was a 7/8 chance that its value would be drawn from the existing population, and a 1/8 chance that a new population would come into effect. Once a new population came into effect, it became the 'existing' population. Each population had its $\mu$ and $\sigma$ drawn randomly from uniform distributions. The 1/8 transition probability, and the other parameters described below, were chosen in order to maximise the dissociation between the perceptual inference variables under study.

## Auditory implementation of algorithm

The algorithm was implemented in the auditory domain, with stimuli taking the form of harmonic complexes, containing only unresolved harmonics (by high-pass filtering from 1.8 kHz). Each harmonic had a random phase offset, which was preserved across all segments, stimuli and subjects. The variable dimension was fundamental frequency (f0; hereafter just 'f' for simplicity), which is the major determinant of perceived pitch. Population μ was limited to the range 120–140 Hz, and σ to the range 1/128–1/16 octaves. Stimulus segments were 300 ms in duration and were smoothly concatenated to avoid any transients at the transitions between segments. This was achieved by defining instantaneous frequency at every point in the stimulus, by calculating the cumulative sum of this, and then by creating harmonics individually in the time domain as follows in Equation 1:

$$a_T = \sin\left(2\pi r + 2\pi h \frac{1}{s} \sum_{t=1}^{T} f_t\right) \tag{1}$$

where $a$ is the amplitude of the waveform, $T$ is the current time point (measured in samples), $t$ is all previous time points, $r$ is the random phase offset for the harmonic, $h$ is the number of the harmonic, $s$ is the sampling rate and $f$ is the instantaneous frequency. This procedure was repeated for every harmonic, from below the high-pass to above the Nyquist frequency. To prevent aliasing, the stimulus was generated at 88.2 kHz sampling rate, then downsampled to 44.1 kHz. The segment duration of 300 ms was chosen as the minimum duration that would capture most of the transient response to the onset of pitch within a stimulus, based on previous work (*Griffiths et al., 2010*). 8000 stimulus segments were presented to each subject, consisting of four blocks of 2000 segments. Blocks were generated by the same rules but were each independently randomly generated. See *Figure 2B* for an example section of the stimuli.

## Subjects and stimulus delivery

Subjects were three patients undergoing invasive electrode monitoring for localisation of medically refractory epilepsy prior to resective surgery. Subjects were not known to have any major cognitive deficits or clinically significant hearing impairment, and none had lesions in the region of auditory cortex. Informed consent for experimentation was obtained from all subjects, and research procedures were approved by the University of Iowa Institutional Review Board. Stimuli were presented diotically, via insert earphones (ER4B; Etymotic Research, Elk Grove Village, IL) through molds fitted to the subject's ear, at the loudest comfortable volume. During the experiments, subjects engaged in an irrelevant auditory task to maintain attention, but a specific performance on this task was not required. This task involved detecting a change to the timbre of individual stimulus segments (64 targets over 8000 segments), which was unrelated to their frequencies or underlying population parameters. Subject 1 performed well on the task, and subjects 2 and 3 performed poorly, with high false alarm rates. The first 100 stimulus segments, and 10 segments following each target and false alarm, were removed from analysis.

## Data acquisition and preprocessing

Recordings were made from one hemisphere in each subject (Subjects 1 and 3: right, Subject 2: left). All subjects had an 8-contact depth electrode placed along the axis of Heschl's gyrus, including anatomically and physiologically defined primary auditory cortex, and a subdural grid overlying superior temporal gyrus. Local field potential data were downsampled to 1 kHz, and electrical noise was filtered out. Time-frequency decomposition was performed with Morlet wavelet convolution, oversampled at 2 Hz frequency resolution and 10 ms time resolution, in the time range -300 to 600 ms from segment onset (i.e. spanning previous, current and subsequent segments) and frequency range 2–100 Hz. The upper frequency bound was limited to 100 Hz in light of previous observations of a lack of qualitative response difference to pitch stimuli between the 80–100 Hz range and higher frequencies (*Griffiths et al., 2010*). The number of cycles per wavelet increased linearly from 1 cycle at 2 Hz to 10 cycles at 100 Hz. The absolute value (i.e. amplitude) of the wavelet coefficients was calculated for artefact rejection purposes, and these were normalised for each frequency (i.e. shifted/scaled to mean 0 and standard deviation 1). For each trial, both the mean (across time and frequency) and maximum normalised amplitude value was recorded, and the frequency histograms of

these were plotted. Thresholds for trial rejection were set, by visual inspection, at the upper limit of the normal distribution of responses, beyond which trials were assumed to contain artefacts. After removal of segments at the start of the experiment, following target segments, and with outlying amplitude values, 89, 86 and 87% of segments remained, for the three subjects, respectively. Data were processed for all electrodes either in Heschl's gyrus or over the superior temporal gyrus. The procedure for selecting electrodes for the final analysis is described later.

## Estimation of perceptual inference variables

Ideal prior predictions, for each stimulus segment were calculated by inverting the algorithm used to generate the frequency values for the stimulus. Human subjects tend to implicitly make near-optimal inferences based on available sensory evidence and past experience (*Ernst and Banks, 2002*; *Körding and Wolpert, 2004*), but in the present study, all that was assumed was an approximate concordance between the subjects' actual inferences and the optimal inferences that we modelled. This seems highly probable given the simplicity of the algorithm and its conformation to Gaussian statistics. Prediction calculation was achieved as follows, using a model inversion scheme illustrated in *Figure 2—figure supplement 1*:

1. A discrete three-dimensional model space was generated (represented as a three-dimensional matrix; *Figure 2—figure supplement 1A*, left), with dimensions corresponding to population μ, population $\sigma$, and f value. Any given value in the matrix indicates P(f|μ,$\sigma$), that is, the probability of a given frequency given a particular $\mu$ and $\sigma$. The columns (all f values for a given $\mu$ and $\sigma$ combination; *Figure 2—figure supplement 1*, upper-right) thus constitute the forward model (by which stimuli are generated), and the planes (all combinations of $\mu$ and $\sigma$ for a given f value; *Figure 2—figure supplement 1*, lower-middle) constitute the inverse model (by which hidden parameters can be estimated from observed f values).
2. 2) For each segment, the model was inverted for its particular f value, yielding a two-dimensional probability distribution for the hidden parameters (*Figure 2—figure supplement 1*, lower-middle). Steps 3-6 were then worked through for each stimulus segment in order, starting at the beginning of the stimulus.
3. These probability distributions, for each segment subsequent to the most recent estimated population change (as defined later), were multiplied together, and scaled to a sum of 1. The resulting probability distribution (*Figure 2—figure supplement 1*, lower-right) thus reflects parameter probabilities taking into account all relevant f values
4. This combined parameter probability distribution was then scalar multiplied with the full model space, in order to weight each of the forward model columns (each corresponding to a particular parameter combination) by the probability of that parameter combination being in effect. The resulting weighted model space was then averaged across parameter dimensions, to yield a one-dimensional (forward) probability distribution, constituting an optimal prediction about the f value of the next stimulus segment, provided a population change did not occur before then. A probability distribution applicable if a population change were to occur was calculated the same way, but without weighting the forward model columns (so as to encompass every possible parameter combination).
5. It was assumed that a population change occurred immediately prior to the first stimulus segment. To infer subsequent population changes, for each segment the probability of observing the present f value was compared for the two probability distributions (the distribution assuming a population change, and the distribution assuming no change), that is, P(f|c) and P(f|~c), respectively, with c denoting a population change. The probabilities were compared, in conjunction with the known prior probability of a population change (1/8), using Bayes' rule, as stated in Equation 2:

$$P(c|f) = \frac{P(f|c)P(c)}{P(f)} \tag{2}$$

Here, P(c|f) is the chance that a population change occurred at that particular time. Given that P(c) is known to be 1/8, and P(f), the total probability of the observed f value, can be rewritten P(f|c)P(c)+P(f|~c)(1-P(c)), the above equation can be rewritten as Equation 3:

$$P(c|f) = \frac{1}{1 + \frac{7P(f|\sim c)}{P(f|c)}} \tag{3}$$

6. For each segment, the above calculation of P(c|f) was made not only with respect to the immediately preceding segment, but also a number of segments preceding that, up to a maximum of 4. Therefore, for segment t, it was possible to conclude that a population change had occurred immediately prior to t, t-1, t-2, t-3, or none of the above. A population change was judged to have occurred at the time point with the highest value of P(c|f), provided this value was greater than 0.5. Using more than 4 lags did not appreciably alter the estimates obtained by model inversion. Importantly, any retrospective inference of population changes did not retrospectively alter any prior predictions generated by the model (e.g. at time t, if a population change were inferred to have occurred at time t-3 then the priors for t-2, t-1 and t were not affected, but only the priors for t+1 onwards).

7. Once the above steps were worked through for each stimulus segment in order, the optimal prior predictions were used to calculate the perceptual inference variables of interest. Predictions themselves were summarised by their mean ($\mu$) and precision (1/variance). Changes to predictions ($\Delta\mu$) were calculated as the absolute change (in octaves) in $\mu$ from one prediction to the next. Surprise (S) was calculated as the negative log probability of the observed f value given the prior prediction, and prediction error (irrespective of prediction precision) was calculated as the absolute difference (in octaves) between the observed f value and the mean of the prior prediction. Mathematically, surprise is directly proportional to prediction precision multiplied by prediction error. Finally, $\Delta f$ was calculated as the absolute difference between the current and preceding value of f.

An example section of the stimulus, along with Bayes optimal prior predictions generated by the above algorithm, are shown in *Figure 2—figure supplement 2*. As the regressors (the variables calculated above) were highly correlated with each other, both instantaneously and over neighbouring segments (see *Figure 2—figure supplement 3*), these were partialised with respect to each of the other regressors, and the preceding and subsequent two values of both the other regressors and themselves. This conservative approach removed a lot of explanatory power from these regressors, but was necessary to be able to uniquely attribute observed neural correlates to a specific process. Importantly, the partialisation did not qualitatively alter the results, except in obscuring the distinction between correlates of S and $\Delta\mu$. Non-partialised results are shown in *Figure 4—figure supplement 3*.

## Correlation analysis

Electrodes were included for further analysis (i.e. included in the averaging process) if they showed a significant response to the stimulus as a whole, based on the single largest response value in time-frequency space. Analysis was based on a 300 ms time window, which was randomly displaced by up to +/- 300 ms for each segment in each permutation (and undisplaced for the actual data) before averaging across segments. The electrodes selected using this procedure are displayed in *Figure 4—figure supplement 5*. In all subjects, electrodes were included from both primary and non-primary auditory cortex.

For each partialised regressor, complex time-frequency data were subject to a two-stage regression approach. First, the complex (i.e. with phase data retained) for every electrode-time-frequency point were regressed against it to yield a pair (real and imaginary) of Pearson product moment correlation coefficients (r). The modulus of these constituted the evoked (time-locked) response. To calculate the induced response, the residuals from this regression (i.e. discarding the evoked component) were converted to amplitude (by taking their modulus), and the regression was performed again. All r values were subjected to a Fisher Z transformation prior to further analysis. Inspection of these responses found no qualitative differences between the responses observed in different divisions of auditory cortex, hence the correlations were averaged over electrodes for further analysis. To quantify the distribution of correlation strengths, the pattern, across time and frequency, of induced correlation coefficients for each regressor was averaged across electrodes and subjects. This pattern was used as a filter, in that it was scalar multiplied with the correlation coefficient pattern for each electrode for each subject, then averaged across time and frequency to yield a single correlation value for that electrode/subject combination. These values, for each regressor, for each subject, were divided by the largest absolute correlation value for that regressor, in order to represent relative correlation strengths on a scale of -1 to 1. These correlation values are displayed in *Figure 4—figure supplement 6*, and show no systematic dissociation between the anatomical distributions of the correlations for the different regressors.

Statistical analyses were all performed using a permutation approach (*Maris and Oostenveld, 2007*), using 100 permutations (in each of which the experimental parameter of interest was removed through randomization, e.g. by shuffling regressor values across stimulus segments) and a significance threshold of $p < 0.05$ corrected. In each permutation, the statistical measure of interest is calculated, and the largest value within each permutation is added to a null distribution, from which a statistical threshold for significance is set. The measure of interest in this approach varied according to the analysis being performed (see below), and included the values at individual time-frequency points averaged across electrodes, mean values across time and frequency at individual electrodes, and mean values across both time-frequency points and electrodes.

To compare surprise and prediction error, for each of these regressors, the mean of the correlation coefficients (across time, frequency, electrode and subject) was calculated within the time window 90–500 ms from segment onset and the frequency window 30- 100 Hz. This was performed once with the regressors partialised as previously described (i.e. for current and adjacent values of $\Delta f$, $\Pi$ and $\Delta \mu$), and again with additional partialisation of each regressor with respect to the other. This latter analysis measures the unique contribution of each regressor in explaining the observed data over and above the other, and was the analysis subjected to statistical analysis.

For the main correlation analysis, significance testing was performed on average correlation values across the three subjects. Points in time-frequency space exceeding the permutation-derived threshold were considered significant. Due to the strong prior hypothesis about gamma oscillations correlating with surprise or prediction error, the statistical analysis was repeated for these regressors but with only frequencies in the gamma range (30–100 Hz) being included in the analysis.

## Acknowledgements

Experiments were conceived and designed by WS, and run by PEG. Electrode implantation was performed by MAH, HK and colleagues. Electrode placements were determined and illustrated by CK and HO. Data analysis was performed by WS. The manuscript was written by WS, with input from all other authors. We thank Professor Karl Friston, University College London, and his research group for helpful comments on the work.

## Additional information

### Funding

| Funder | Grant reference number | Author |
| --- | --- | --- |
| Medical Research Council | MR/J011207/1 | William Sedley |
| National Institutes of Health | NIH R01-DC04290 | Christopher K Kovach<br>Hiroyuki Oya<br>Hiroto Kawasaki<br>Matthew A Howard III |
| Wellcome Trust | WT091681MA | Phillip E Gander<br>Timothy D Griffiths |

The funders had no role in study design, data collection and interpretation, or the decision to submit the work for publication.

### Author contributions

WS, Conception and design, Analysis and interpretation of data, Drafting or revising the article; PEG, CKK, HO, HK, MAH, Conception and design, Acquisition of data, Drafting or revising the article; SK, TDG, Conception and design, Drafting or revising the article

### Author ORCIDs

William Sedley, http://orcid.org/0000-0002-9371-0962

### Ethics

Human subjects: The study was approved by the University of Iowa Institutional Review Board, and with full informed written consent from all participants.

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
