## [Decision Letter]

Thank you for submitting your work entitled "Neural Signatures of Perceptual Inference" for consideration by *eLife*. Your article has been reviewed by three peer reviewers, and the evaluation has been overseen by Andrew King as the Reviewing Editor and Eve Marder as the Senior Editor.

The following individuals involved in review of your submission have agreed to reveal their identity: Jonas Obleser, Luc Arnal and Peter Kok (peer reviewers).

The reviewers have discussed the reviews with one another and the Reviewing Editor has drafted this decision to help you prepare a revised submission.

Summary:

There is general agreement among the reviewers that this is a carefully performed and potentially influential study of the neural correlates of predictive coding. In particular, the use of human ECoG recordings from human auditory cortex to investigate the local field potential oscillatory signatures of change to prediction, precision of predictions and surprise represents a novel and important extension of previous work in this field.

Essential revisions:

Although they are generally enthusiastic about this paper, the reviewers have raised a number of issues that will need to be addressed, most of which relate to statistical/methodological issues, the organization of the manuscript, and to acknowledgement of prior work.

1) The reviewers have pointed out that your claim (Results/Discussion, fourth paragraph) that this study provides the first direct evidence that beta oscillations are involved in updating the content of sensory predictions is incorrect. Previous work on the role of beta oscillations has not been cited: Buschman & Miller (2007) Science 315:1860-1862; Bastos et al., 2014, first published on BioRvix, and subsequently in Neuron (2015) 85:390-401. Similarly, in the second paragraph of the Introduction, it is stated that "A neural correlate of precision.… has not been proposed." Some proposals about the neural implementation of precision have in fact been made in previous studies; see e.g. Feldman & Friston (2010) Front Hum Neurosci 4:215, and Kanai et al. (2015) Philos trans R Soc Lond B 370(1668).

2) The reviewers (and editors) found it difficult for the reader to navigate through the many (main and supplementary) figures. We suggest that the authors carefully consider some reorganising of the figures and the way the data are presented, as this will likely considerably improve the manuscript. Please take full advantage of *eLife* formatting of figures and figure supplements.

3) Clarification is needed about when single subject data are being described and when data from the three participants has been combined. While it is crucial to show that your observations are reliable across participants, it is probably not necessary to show single subject response profiles on main figures. You should consider showing the most relevant effects at the group level on main figures while providing single subject effects in supplements to the figures. This would also allow grouping the main findings of each figure into a single one, thereby providing a more synthetic view of the results.

4) You show that the regressors are highly correlated with each other (e.g. prediction error and surprise). While this validates the authors' partialisation approach, the reviewers are concerned that you only show the (tiny) portion of the data that is not shared between regressors. In addition to showing what is specific to each regressor, it would be useful for the reader to know what is shared between those regressors, i.e. in which parts of brain responses reflect common processes.

5) Where average r values are further processed, please confirm that this was only done after prior Fisher z transformation.

6) From point 7 of the explanation of the prediction calculation (subsection “Estimation of perceptual inference variables”), it seems that first population change positions were estimated, before estimating optimal predictions and the other perceptual inference variables. In point 6, it is explained that in order to estimate population change positions, segments up to three trials back were considered, and inferred population changes could be assigned, retroactively, to up to three trials back. Does this mean that the other perceptual inference variables could also be changed retroactively? In other words, when at trial n it was inferred that a population change occurred at trial n-3, were the prediction, precision, surprise and prediction error for trial n-3 affected by this? It seems implausible for a neural surprise response on trial n-3 to be affected by a prediction made during trial n.

7) The reviewers agree that a strong case has been made to link beta and gamma oscillations in auditory cortex to the updating of and the error in predictions, respectively, but are less convinced about the claims made in respect to alpha (or rather alpha-delta) oscillations. The precision effect reported in the alpha band seems rather weak and is hardly visible in Figure 4. Again, one way to show this more convincingly would be to show the time-frequency map of these effects averaged at the group level and not only the significance. Although appropriately corrected statistics for multiple comparisons are used, it would be more convincing to provide additional information and analyses to backup this finding and support the claim that alpha oscillations encode precision. Otherwise, you should be more circumspect in your conclusions about this link.

---

## [Author Response]

*Essential revisions: Although they are generally enthusiastic about this paper, the reviewers have raised a number of issues that will need to be addressed, most of which relate to statistical/methodological issues, the organization of the manuscript, and to acknowledgement of prior work. 1) The reviewers have pointed out that your claim (Results/Discussion, fourth paragraph) that this study provides the first direct evidence that beta oscillations are involved in updating the content of sensory predictions is incorrect. Previous work on the role of beta oscillations has not been cited: Buschman & Miller (2007) Science 315:1860-1862; Bastos et al., 2014, first published on BioRvix, and subsequently in Neuron (2015) 85:390-401.*

We realise that, in attempt to maintain brevity of the manuscript, we have not cited all relevant work on the subject. However, I think any disagreement here is over the definition of the term ‘direct evidence’. We are aware of compelling evidence that there is a spectral asymmetry inherent in hierarchical neuronal communication, with beta being a major feedback frequency (as demonstrated in the two studies mentioned above, and others), and that in predictive coding feedback connections are argued to convey predictions, or updates to predictions. However, we have not been aware of any evidence specifically and quantitatively linking beta oscillations to prediction updates as we have demonstrated in the present work. We therefore would argue that what has been made previously is a strong case on circumstantial evidence, for which our study is the first to provide direct confirmation. Specifically, our understanding of the above two studies is that Buschman and Miller found low-frequency feedback connections from prefrontal cortex to parietal cortex to be relatively selective for top-down attentional control, and that Bastos found asymmetrical spectral Granger causality, with feedback being associated with beta band connections. The text has been updated as follows:

The second introductory paragraph now reads:

“The layer-specific separation of higher and lower frequency oscillations (Spaak et al., 2012), and the forward/backward asymmetry of high/low frequency oscillations (Buschman and Miller, 2007; Fontolan et al., 2014; van Kerkoerle et al., 2014, Bastos et al., 2015), are supported by direct evidence. […] While there is a strong case that beta oscillations are involved in top-down neural communication, evidence specifically linking beta oscillations to predictions is presently limited and indirect (Arnal and Giraud, 2012), but includes observations that there is interdependence of gamma and subsequent beta activity in both in-vivo (Haenschel et al., 2000) and in-silico (Kopell et al., 2011) studies, and that omissions of expected stimuli induce a beta rebound response (Fujioka et al., 2009).”

*Similarly, in the second paragraph of the Introduction, it is stated that "A neural correlate of precision.… has not been proposed." Some proposals about the neural implementation of precision have in fact been made in previous studies; see e.g. Feldman & Friston (2010) Front Hum Neurosci 4:215, and Kanai et al. (2015) Philos trans R Soc Lond B 370(1668).*

We acknowledge our mistake here. What we intended to say was that an oscillatory code for precision of predictions has not been proposed. We are aware that cholinergic tone is a likely neurochemical/neuromodulatory correlate of precision, and that gamma synchrony likely reflects precision-weighted prediction errors (as are discussed in Feldman and Friston 2010). Kanai et al. 2015 also describes a neuromodulatory, not oscillatory, mechanism of precision control. The text has been updated with these citations, and now reads:

“An oscillatory correlate of precision, to our knowledge, has not been proposed, though precision might affect the magnitude of gamma responses to prediction violations (Feldman and Friston, 2010). While an oscillatory correlate is possible, a case has been made that neuromodulatory connections alone, for instance from the basal forebrain cholinergic system, may be sufficient to dynamically mediate precision in sensory hierarchies (Feldman and Friston, 2010; Kanai et al., 2015).”

*2) The reviewers (and editors) found it difficult for the reader to navigate through the many (main and supplementary) figures. We suggest that the authors carefully consider some reorganising of the figures and the way the data are presented, as this will likely considerably improve the manuscript. Please take full advantage of eLife formatting of figures and figure supplements.*

The figures have now been simplified and reorganised, in keeping with *eLife*’s particular conventions. Most importantly, composite supplemental figures have been broken down into multiple sub-figures for clarity.

*3) Clarification is needed about when single subject data are being described and when data from the three participants has been combined. While it is crucial to show that your observations are reliable across participants, it is probably not necessary to show single subject response profiles on main figures. You should consider showing the most relevant effects at the group level on main figures while providing single subject effects in supplements to the figures. This would also allow grouping the main findings of each figure into a single one, thereby providing a more synthetic view of the results.*

This is a good suggestion, which we have followed. The main Figure 4 now shows group-level induced and evoked results, and both single subject data and alternative analyses are shown in supplemental. We have been through the text to ensure that it is clear what type of data are being referred to.

*4) You show that the regressors are highly correlated with each other (e.g. prediction error and surprise). While this validates the authors' partialisation approach, the reviewers are concerned that you only show the (tiny) portion of the data that is not shared between regressors. In addition to showing what is specific to each regressor, it would be useful for the reader to know what is shared between those regressors, i.e. in which parts of brain responses reflect common processes.*

While we had previously shown a supplemental figure with non-partialised regressors (which shows the data explained by the commonalities between the regressors as well as the unique contributions), we have expanded on this in the relevant figure, and also included a figure mapping the percentage of variance, across time-frequency space, explained by the whole model. We believe that with these figures included, the reader is able to appreciate the unique regressor correlates in the context of all correlates of the model. The relevant parts of the text have been modified as follows:

“Furthermore, the LFP data variance explained by the entire model (Figure 4—figure supplement 1) was around 1%.”

“We therefore repeated the analyses with only the contemporaneous value of DELTAf being partialled out (Figure 4—figure supplement 3). This analysis found highly significant correlates of *precision* (Π) in the full delta-alpha range, spanning the previous, current and subsequent segments, but we cannot attribute the delta-theta component to Π with absolute confidence.”

*5) Where average r values are further processed, please confirm that this was only done after prior Fisher z transformation.*

We had not done this previously. Analyses have now been re-run with a Fisher Z transformation being applied at each time-frequency-channel-subject combination prior to any averaging. We note that, as the r values in question are very small, the effect of the Fisher Z transformation is negligible.

*6) From point 7 of the explanation of the prediction calculation (subsection “Estimation of perceptual inference variables”), it seems that first population change positions were estimated, before estimating optimal predictions and the other perceptual inference variables. In point 6, it is explained that in order to estimate population change positions, segments up to three trials back were considered, and inferred population changes could be assigned, retroactively, to up to three trials back. Does this mean that the other perceptual inference variables could also be changed retroactively? In other words, when at trial n it was inferred that a population change occurred at trial n-3, were the prediction, precision, surprise and prediction error for trial n-3 affected by this? It seems implausible for a neural surprise response on trial n-3 to be affected by a prediction made during trial n.*

This seems to be an issue with our explanation rather than the method, and I realise my mistake in how I described things. To clarify: the algorithm was only allowed to work in ways plausible for an ideal biological observer. The starting assumption was that a population change definitely occurred at trial zero. Then, at each segment subsequently, the algorithm inverted all segments since the last inferred population change (initially trial zero) and decided whether (and if so, where) a population change was likely to have occurred since then. Once this was decided, priors for the upcoming stimulus were generated, and these were ‘set in stone’ (i.e. could never be modified after this). Then, the algorithm would move onto the next trial. Put another way, any retrospective analysis in the algorithm was limited to the inference of population changes, and did not apply to the generation of priors. The text has now been modified as follows:

“Therefore, for segment t, it was possible to conclude that a population change had occurred immediately prior to t, t-1, t-2, t-3, or none of the above. […] Once the above steps were worked through for each stimulus segment in order, the optimal prior predictions were used to calculate the perceptual inference variables of interest.”

*7) The reviewers agree that a strong case has been made to link beta and gamma oscillations in auditory cortex to the updating of and the error in predictions, respectively, but are less convinced about the claims made in respect to alpha (or rather alpha-delta) oscillations. The precision effect reported in the alpha band seems rather weak and is hardly visible in Figure 4. Again, one way to show this more convincingly would be to show the time-frequency map of these effects averaged at the group level and not only the significance. Although appropriately corrected statistics for multiple comparisons are used, it would be more convincing to provide additional information and analyses to backup this finding and support the claim that alpha oscillations encode precision. Otherwise, you should be more circumspect in your conclusions about this link.*

We think this is a fair point. We have followed the suggestions above, in displaying the averaged group data. We also note that the appropriate correction for multiple comparisons does mean that the finding is almost certainly genuine, however small the time-frequency area concerned is (particularly as such extensive partialisation has been performed as a necessary preprocessing step). We also, however, respect the fact that this is an entirely new concept that does not build upon a robust background literature (as the other main findings do), and therefore agree we should be more circumspect in our interpretation. We do note, though, that it is not merely of interest that precision appears to have its own oscillatory correlate, but also that the inclusion of precision in the analysis at all (in terms of it being partialled out from the other regressors) produces results that are more in keeping with prior theory; with precision included, surprise correlates solely with gamma oscillations, whereas with precision omitted surprise shows a strong late negative low-frequency correlation which is harder to reconcile with existing theory. Overall, we feel genuinely unsure whether the precision findings will prove hugely important across neuroscience, or an alternative explanation will be found in due course. With this in mind, we have revised the text as follows.

The Abstract now reads:

“… and discovered, using direct recordings from human auditory cortex, that *surprise* due to prediction violations is encoded by local field potential oscillations in the gamma band (>30 Hz), changes to *predictions* in the beta band (12-30 Hz), and that the *precision* of predictions appears to quantitatively relate to alpha band oscillations (8-12 Hz).”

The Results section now reads:

“Prediction *precision* (Π) correlated positively with delta-alpha (2-12 Hz) frequency oscillations (for the whole 0-300 ms period from segment onset), though this was only significant in the alpha frequency range, and fell slightly below significance in the delta-theta range. […] However, given its sole significant association in the main analysis (Figure 4) is with Π, and that a low-frequency correlate of S is not expected based on prior literature, we favour the interpretation that low-frequency oscillations are a correlate of the precision of prior predictions.”

“We have also found that the precision of predictions is correlated to the magnitude of alpha oscillations, and possibly delta and/or theta also, thereby raising the possibility of an oscillatory mechanism for the control of precision.”

The relevant paragraph concludes with:

“While an oscillatory code for precision could have far-reaching implications, we must respect the fact that this is a novel finding and thus requires corroboration from additional studies with alternative methodology.”